# Assessment of Tilapia (*Oreochromis* spp.) Welfare in the Semi-Intensive and Intensive Culture Systems in Thailand

**DOI:** 10.3390/ani13152498

**Published:** 2023-08-02

**Authors:** Tuchakorn Lertwanakarn, Thitima Purimayata, Thnapol Luengyosluechakul, Pau Badia Grimalt, Ana Silvia Pedrazzani, Murilo Henrique Quintiliano, Win Surachetpong

**Affiliations:** 1Department of Veterinary Physiology, Faculty of Veterinary Medicine, Kasetsart University, Bangkok 10900, Thailand; tuchakorn.l@ku.th; 2Graduate Program in Animal Health and Biomedical Science, Faculty of Veterinary Medicine, Kasetsart University, Bangkok 10900, Thailand; purimayata.thitima@gmail.com (T.P.); kothnapol@gmail.com (T.L.); 3Department of Veterinary Microbiology and Immunology, Faculty of Veterinary Medicine, Kasetsart University, Bangkok 10900, Thailand; 4FAI Registered Office Company Address, The Barn, Wytham, Oxford OX2 8QJ, UK; paubadiagrimalt@gmail.com; 5Wai Ora Aquaculture and Environmental Technology Ltd., Curitiba 80240-050, Brazil; anasilviap@ufpr.br; 6FAI Farms, Londrina 86115-000, Brazil; murilo.quintiliano@faifarms.com

**Keywords:** tilapia farming, welfare assessment, fish welfare, earthen pond, cage culture

## Abstract

**Simple Summary:**

In this study, we evaluated welfare indicators at eight small-scale semi-intensive and intensive tilapia farms in Thailand, with a focus on four welfare categories: health, environment, behaviour, and nutrition. The results showed differences between the relative scores across all four welfare-indicator categories. Specifically, the behavioural assessments revealed poor welfare practices during the catching process on the tilapia farms, while the nutritional assessments showed differences in the feed conversion rates and K factor values. We identified correlations between the nutritional, environmental, and health indicators, which provided information about the critical welfare points in fish farming. By understanding welfare indicators and improving farming practices, farmers can produce healthier fish and improve their profits and the quality of the fish supplied to consumers. Taken together, our study provides valuable information that can be applied to the broader context of sustainable aquaculture and animal welfare.

**Abstract:**

Welfare assessments have risen to prominence in the aquaculture industry, with increasing awareness of their significance among stakeholders in Thailand. In this study, we conducted a welfare assessment of tilapia (*Oreochromis* spp.) farms in Thailand, focusing on health, environmental, behavioural, and nutritional indicators. Comparing semi-intensive (earthen ponds) and intensive farming practices (cage culture), we found significant differences in the overall health score, particularly at farm F due to a disease outbreak (Kruskal–Wallis, *p* = 0.01). Skin and fin scores varied across farms, indicating their potential as indicators of tilapia health. Environmental assessments revealed differences in transparency between the two culturing systems (Mann–Whitney, *p* = 0.02). During the harvesting process, tilapia behaviours indicated poor welfare across all farms. However, no statistically significant difference in overall welfare scores was found between the two culturing systems. Correlations were observed between nutritional, environmental, and health indicators, with negative correlations between fish density and water transparency (*r* = −0.87, *p* = 0.02), presence of inhabitants (*r* = −0.78, *p* = 0.04), feeding behaviours (*r* = −0.78, *p* = 0.04), and swimming behaviours during capture (*r* = −0.98, *p* = 0.001). These findings provide valuable insights to enhance tilapia-farming practices and welfare in Thailand.

## 1. Introduction

Tilapia (*Oreochromis* spp.) farming is a significant industry that comprises approximately 9% of all finfish in the global market [1]. In Thailand, tilapia is the most economically important freshwater fish, with annual production volumes of over 210,000 metric tons and a market value exceeding USD 300 million [2]. However, the rapid expansion of tilapia aquaculture, particularly in intensive culture systems, has raised concerns about potential negative impacts on fish health, water quality, sustainability, and animal welfare. The application of animal welfare principles in the production of both terrestrial and aquatic animals is becoming increasingly important in meeting consumer demand for safe food and ensuring the production of healthy animals and high-quality products [3].

While the concept of animal welfare is considered a small part of good aquaculture management practices, including Good Aquaculture Practice (GAP) [4], the Aquaculture Stewardship Program [5], and Global Seafoods Alliance Best Aquaculture Practice [6], the importance of animal welfare remains a concern of the World Organization of Animal Health [7]. Indeed, various welfare assessment frameworks designed specifically for aquaculture practices, such as SWIM [8], FISHWELL [9], and MyFishCheck [10], have been developed. These concentrate solely on animal welfare and incorporate a range of indicators to determine an overall welfare score based on scientific evaluations. Interestingly, these principles for assessing animal welfare have been applied in economically important fish species such as salmon (*Salmo salar*), trout (*Oncorhynchus mykiss*), and pike perch (*Sander lucioperca*) [8,10]. 

In tilapia, it is essential to incorporate welfare considerations into an assessment protocol to address welfare concerns during tilapia production. In 2022, Flores-García and colleagues evaluated the effects of feeding rates and several welfare indicators to aid in decision making regarding and the evaluation of tilapia production through a color-coding approach inspired by traffic signals and assigned scores to the colours red, yellow, and green [11]. Another study in Brazil revealed that the application of welfare assessments in tilapia can promote sustainable and ethical aquaculture practices that benefit fish health and growth. These assessments included four categories of indicators: health, environment, behaviour, and nutrition [12].

Notwithstanding, relatively few studies have been conducted on welfare assessment indicators for tilapia and their applicability to tilapia farms. Such indicators may also not be applicable to tilapia-farming practices in different countries due to the variabilities in farming practices. The objective of this study was therefore to apply welfare assessment indicators to evaluate tilapia-farming practices in Thailand. These indicators, which comprised the four categories of health, environment, behaviour, and nutrition, were based on the animal freedoms delineated by the Farm Animal Welfare Committee [13] and were used to assess tilapia farms with semi-intensive and intensive culture systems. The study included a constructive analysis and applied the principles of animal welfare to improve tilapia-farming practices in Thailand. Furthermore, our approach can be used to assess animal welfare on tilapia farms in other countries.

## 2. Materials and Methods

### 2.1. Tilapia Farms, Data Collection, and Ethics

We assessed tilapia welfare at eight tilapia farms; five of these farms had earthen ponds, and three used the cage system. The farms were located in different regions of Thailand, with three farms in northern Thailand, one in the central region, two in the east, and two in the west (Figure 1). The data were collected through direct interviews and observations at the farms. All the welfare criteria utilised in this study were modified from the previous protocol for tilapia farms [12]. The participating farms were all informed about and consented to the disclosure of their information [4]. This project was approved by the Animal Use Ethics Committee (ACKU66-VET-010) of Kasetsart University.

### 2.2. Welfare Criteria

Using the four welfare categories, the health status of the fish was physically assessed via direct observation of the fish’s eyes, jaws, opercula, skin, fins, gills, and spines as well as ectoparasitic infection and reported in distribution percentages, as demonstrated in Appendix A. The sizes of the sampled fish, which included the length (cm) and weight (g), were measured using meter tape and a digital scale. A relative score for each parameter was also computed using the following equation:$$Relative\;score=\frac{\left(1\times n1\right)+\left(2\times n2\right)+(3\times n3)+(4\times n4)\;}{n1+n2+n3+n4}$$
where n1, n2, n3, and n4 represent the number of fish observed in the score groups 1, 2, 3, and 4, respectively. In accordance with the equation, a higher score indicated poorer tilapia health conditions at the respective farms, as previously described.

The assessed environmental indicators comprised physicochemical water quality parameters and included temperature, pH, transparency, dissolved oxygen (DO), non-ionised ammonia, nitrites, and alkalinity (Appendix A). The DO and temperature were determined using commercial probes (YSI Pro20, Yellow Springs, OH, USA), and the pH of the water was measured using a tester probe (multifunction model number C-600, Water Tools, Qingdao, China). A Secchi disk was used to determine water transparency, while the nitrites, alkalinity, and total ammonia nitrogen (TAN) were measured using specific test kits (Monitor^®^, Bangkok, Thailand). The non-ionised ammonia (NH_3_) was calculated based on the TAN under specific temperatures and pH levels. Predators and other inhabitants were evaluated by the investigator. The stocking densities of the earthen ponds and cages were determined using the farm records.

The behaviour of the tilapia during feeding and capture were graded as shown in Appendix A. For nutritional status, the fish’s weight, age, stocking density, type of feed, and percentage of crude protein (CP%) were considered, and the staff of each farm were interviewed to obtain the feed information (Appendix A). The feed conversion ratio (FCR) was calculated from the total feed consumed (kg) per live weight gain of fish (kg), and the condition factor (K) was calculated as the percentage of weight (g) divided by the cubed power of the fish length (cm). Values within the optimal limits were graded as score 1, whereas scores 2, 3, and 4 were allocated for values that were 10%, 20%, and over 20% of the normal range, respectively. 

### 2.3. Statistical Analysis

The obtained welfare parameters are described as values and scores. The non-parametric scores between the farms and parameters were compared using the Kruskal–Wallis test. The comparison of two production systems were evaluated using the Mann–Whitney test. Spearman’s correlation was analysed to identify the relationships between the health, environmental, behavioural, and nutritional indicators. *p*-values <0.05 were considered statistically significant. All the statistical analyses were performed using GraphPad^®^ Prism software, version 9.0 (San Diego, CA, USA)

## 3. Results

### 3.1. Demographic Information of the Studied Tilapia Farms

The information regarding the tilapia farms included in this study is provided in Table 1. The majority of the fish in each earthen pond were Nile tilapia, whereas all the cage systems cultivated red tilapia. The earthen ponds had larger areas, which ranged from 3200 to 28,800 m^2^, compared to the cage systems at 20–25 m^2^. The fish at each farm ranged in age from 75 to 243 days and had an average weight of 106.6–710.6 g. No significant differences were observed in terms of fish age and weight between the two culturing systems chosen for this study.

### 3.2. Welfare Indicators 

#### 3.2.1. Health Indicators

In this study, we assessed the health status of the tilapia using various indicators, which included the eyes, jaws, opercula, skin, fins, gills, spine, ectoparasites, and mortality rates. The results are presented in Table 2 and Table 3 and Appendix A as distribution percentages and relative scores. For farm A, over 95% of the tilapia were graded with a score of 1 for eyes, jaws, opercula, and spine, which indicated good relative scores (1.00–1.03). However, a larger proportion of the fish were graded as 2 or 3 for skin, fins, and gills, which denoted poorer relative health scores (1.47, 1.21, and 1.35, respectively). Similar results were obtained for farms B, C, G, and H, where the majority of the tilapia were graded with a score of 1 for all the health indicators. A small proportion of the fish were graded 2 and 3, but none were graded 4, resulting in relative scores of 1.00–1.40. In contrast, farms D and E had a higher number of fish graded 2 or 3 for skin, gills, and fins, which resulted in higher relative health scores (1.26–1.53). Nevertheless, the eyes, jaws, opercula, and spines of most of the tilapia on these farms were healthy, with relative scores ranging from 1 to 1.1. Notably, farm F had the worst health scores, where only half the fish were graded with a score of 1 for each indicator, which resulted in relative scores of over 2.00. However, this was the only farm where ectoparasites were detected in 60% of the assessed fish, and it had the highest mortality rate (50%, score 3). Overall, farm F had significantly worse health scores than the other farms (Kruskal–Wallis, *p* = 0.01). Although the mortality rates were significantly higher in the cage culturing system than in the earthen ponds (Table 2), no differences were evident in the overall relative health scores. Importantly, the relative scores for skin and fins of all the farms were significantly different from those of the other indicators.

#### 3.2.2. Environmental Indicators

The recorded values and scores for the environmental indicators are presented in Table 4. At farm A, the water temperature, pH, NH_3_, and NO_2_^−^ were within acceptable levels. However, high levels of DO (8.25 mg/L, score 4) and alkalinity (102 mg/L, score 2) were observed. Of note, the tilapia at this farm were co-cultivated with white shrimp and graded with a score of 3. For farms B, D, F, G and H, most of the water quality parameters were graded 1, with mild reductions in the DO at farm G (5.2 mg/L, score 2) and mild elevations in alkalinity at farms D (102 mg/L, score 2), F (119 mg/L, score 2), and G (119 mg/L, score 2), respectively. At farm C, optimal values were observed for the water temperature, NH_3_, NO_2_^−^, and alkalinity; however, the water pH (8.7, score 2) and DO (8.9 mg/L, score 4) were measured as high. The water temperature, pH, NO_2_^−^, and alkalinity at farm E were within the preference range; however, the measurements revealed low oxygen saturation (3.4 mg/L, score 3) and a high level of NH_3_ (0.08 mg/L, score 2). Notably, the tilapia at farm E were co-cultivated with white shrimps and catfish. All the farms had good fish densities in the ponds, but uncontrolled populations of predators, such as birds and common silver barbs (*Barbonymus gonionotus*), were observed at all the farms. A comparison of the two production systems showed that earthen ponds had a significantly higher score for water transparency than did cage cultivation (Mann–Whitney, *p* = 0.02). However, the overall environmental scores were in a similar range for both farming practices.

#### 3.2.3. Behavioural Indicators

The behavioural assessments, which included behaviours during feeding and the capturing process, were evaluated at all eight farms (Table 5). We observed that the feeding patterns were influenced by several factors, including pond size, feed type, fish health, water quality, and the experience of the farmers. At farm A, the feeding area measured 5 × 5 × 2 m^3^ with a narrow boundary, which resulted in a high level of competition among a large number of fish for feed. Consequently, the fish took more than 6 min (score 4) to finish feeding. At farms B, C, and D, on the other hand, 6–9 feeding areas were available, which allowed the fish to finish feeding within 6 min (score 1). Interestingly, the owner of farm E fed the fish manually using a boat, and the fish spent more than 6 min (score 4) consuming food. Unlike those in the earthen ponds, the fish in all the cage systems took around 5–6 min to finish feeding (score 1). In terms of the catching process, most of the fish were found swimming in different directions and/or exhibiting decreased activity. They were piled against the nets, floated on their sides, showed signs of exhaustion with prolonged body exposure to air, and were then sorted into baskets for a while (score 4). Farm H demonstrated good management in terms of feeding and capturing scores. No significant differences were found in the behaviour of the fish cultivated in earthen ponds or cages.

#### 3.2.4. Nutritional Indicators

The nutritional status of the fish at the different farms was assessed based on the feed type, crude protein percentage (CP%), FCR, and K factor values (Table 6). The tilapia at farms A, B, C, D, F, G, and H were provided with commercial diets that contained a crude protein ratio of 25–32% and received regular and consistent feeding. These farms exhibited good FCRs with values ranging from 1.1 to 1.3 (score 1). Farms F and G had slightly higher FCR values (1.52 and 1.41, respectively; score 2). However, the K-values, which were randomly computed from grow-out fish, indicated sub-standard values at all the farms (2.2–3.7), with scores ranging from 2 to 3. At farm E, on the other hand, a homemade diet with an unknown CP% was used, and the FCR could therefore not be calculated. No differences were observed between the two types of culturing systems.

All the farms underwent assessments with 25 welfare indicators (23 indicators for farm E), which included health, environmental, behavioural, and nutritional aspects (Figure 2). Despite farm F obtaining higher scores in the health category, no significant differences were found in the overall welfare scores across all the farms (*p* = 0.16). Additionally, the total welfare scores for the earthen ponds and cage systems were found to be similar.

### 3.3. Correlation between Nutritional, Environmental, and Health Indicators

To identify possible correlations between the environmental, behavioural, and nutritional indicators, we conducted a correlation test (Figure 3) and found positive correlations between the health scores, such as jaw and fin lesions (*r* = 0.85, *p* = 0.01), skin and gills (*r* = 0.79, *p* = 0.02), and fins and gills (*r* = 0.74, *p* = 0.04). Importantly, the mortality rates were significantly correlated with eyes (*r* = 0.82, *p* = 0.04) and jaw lesions (*r* = 0.78, *p* = 0.03). Negative correlations between fish density and water transparency (*r* = −0.87, *p* = 0.02), the presence of inhabitants (*r* = −0.78, *p* = 0.04), feeding behaviours (*r* = −0.78, *p* = 0.04), and swimming behaviours during capture (*r* = −0.98, *p* = 0.001) were also observed. We further noticed a correlation between feeding behaviours and the presence of inhabitants (*r* = −0.78, *p* = 0.04) and capturing behaviours and water transparency (*r* = 0.88, *p* = 0.02).

## 4. Discussion

Due to the increasing demand for aquatic animal products, the aquaculture industry has undergone significant growth in recent decades. This expansion has brought about a greater need to prioritise the health and welfare of aquatic animals in intensive farming practices. Tilapia, a valuable aquatic species globally, requires proper welfare management to ensure the success and sustainability of tilapia farms. However, despite the availability of welfare assessment methods for various fish species, research on tilapia welfare has been limited. In this study, we employed a four-category welfare assessment method previously used in Brazil to evaluate the welfare of tilapia farms in Thailand. Our findings revealed no significant differences in the overall welfare of the tilapia raised in earthen ponds and cages. The welfare scores reflected the quality of the practices applied in the management of Nile tilapia and red tilapia farms in Thailand. Specifically, the farms that were well managed and had healthy fish tended to have better scores compared to the farms with poor farm management practices and where fish health issues were experienced. The details of the study results and various criteria were categorised into the four key welfare assessment areas.

### 4.1. Health Indicators

The assessment of external lesions has been proposed as a potential indicator of fish welfare in fish aquaculture. Various factors such as disease, improper handling, and stress from poor environmental conditions contribute to external damage and lesions in fish. The presence and severity of these injuries can indicate inadequate welfare practices and husbandry. Our findings identified tilapia fins and skin as the two primary indicators of external injuries, which is consistent with previous welfare assessments at tilapia farms in Brazil and Mexico [11,12], as well as in other fish species including rainbow trout and salmon [8]. It is crucial to determine the underlying cause of injuries and lesions, such as inappropriate management, as these can indicate poor animal welfare and lead to poor fish health. Although no correlations were found between fish health and the other indicators in this study, we observed injuries on the skin, fins, and gills of some fish at farms A, B, C, D, E, G, and H, which were caused by random sampling with nets. This highlights the need for improved handling management at tilapia farms. During the welfare assessment at farm F, most of the fish exhibited poor overall health scores due to coinfection by ectoparasites, tilapia lake virus (TiLV), and bacterial infections. Previous studies have demonstrated that tilapia may develop severe lesions in various organs when simultaneously infected with bacteria, ectoparasites, and TiLV [14,15]. Accordingly, when assessing the welfare status of fish, it is imperative to consider the disease status and specific lesions caused by each pathogen or a combination thereof. Indeed, these lesions differ from external lesions caused by improper handling or sampling of fish. 

The results of this study revealed that the tilapia raised in cages had higher mortality rates compared to those raised in earthen ponds, which is consistent with previous findings regarding river-based cage culture practices [16,17]. Factors such as continuous exposure to pathogens, rapid changes in water quality, and biosecurity limitations may have contributed to the higher mortality rates observed in the cages than the earthen ponds in our study. We also found a correlation between the mortality rates and alterations in the eyes and jaws, such as swollen eyes and abscesses on the jaw, which are commonly observed in tilapia infected with TiLV and *Streptococcus* spp. [16,18]. While farms F and G exhibited higher fish mortality rates than the other farms, no significant correlations were found between the mortality rates and other factors, except for higher stocking densities and alkalinity levels at these two farms. Researchers who have conducted similar studies at tilapia farms in other countries have reported correlations between high stocking densities and stress, mortality rate, and disease outbreaks [19,20,21]. Beyond the differences in rearing methods involving earthen ponds and floating cages, the species of fish, notably Nile tilapia and red hybrid tilapia, may also impact welfare assessments. For example, red hybrid tilapia, while having high market value and popularity in local markets, tend to be more susceptible to diseases compared to Nile tilapia. Overall, the evaluation of external lesions and health indicators has the potential to serve as a tool for monitoring and improving fish welfare at tilapia farms. 

### 4.2. Environmental Indicators

Like other aquatic species, tilapia are susceptible to changes in water quality, temperature, and the availability of oxygen in water. Environmental stressors such as overcrowding, high stocking densities, and poor water quality can cause significant physiological and behavioural alterations in tilapia, which results in their increased susceptibility to diseases as well as reduced growth and survival rates [22,23]. Regular monitoring of the quantity and quality of water is therefore important during welfare assessments. All the tilapia farms in this study were located near natural water resources that were irrigated by channels from rivers. Only farm A was able to store runoff water in empty ponds, while the other farms had limited space for water storage and relied solely on natural sources for their water supplies. The overall environmental scores did not differ significantly between the different farms and the types of cultivation (i.e., ponds vs. cages). Notwithstanding, the earthen ponds showed significantly more water turbidity than the cages, which could be attributed to the addition of microalgae or phytoplankton to the water to enrich the water with nutrients and promote the gut performance of the fish [24]. Although these plankton and microbiomes increase natural protein sources for fish, it is important to consider the potential risks of unwanted flavours in fish flesh provoked by compounds such as 2-methylisoborneol, which are produced by blue-green algae [25]. We identified alkalinity as the most abnormal finding, and this was reported for farms A, D, F, and G. Although it did not show a significant correlation with the other indicators, these farms were associated with high mortality rates. Notably, alkalinity is the measurement of the carbonates and bicarbonates in water, and these compounds affect the growth performance of fish [26,27]. The discrepancy pattern between alkalinity and pH found in this study may have been influenced by alterations in the total hardness and/or salinity of the water, as previously described [26]. The measurement of water hardness and salinity, together with alkalinity, should therefore be included in welfare assessments for tilapia, as advised for other species [8,9,28]. Apart from alkalinity, deviations in DO levels were also reported for farms A, C, E, and G, with no specific correlations between this parameter and others. The high DO levels at farms A, C, and G may not have affected the fish performance, as tilapia can tolerate oxygen saturation of up to 40 mg/L [29,30]. However, low oxygen (3.4 mg/L) was evident at farm E despite the presence of aeration. We speculated that the addition of manure to the water may have influenced this parameter and increased the NH_3_ levels in the water. Additionally, at a low pH, the NH_3_ would have turned into an ammonium (NH_4_^+^) form, which is less toxic to fish [31]. Although the water temperature recordings in our study were within the normal range for all the farms, a single measurement may not be representative of welfare analyses. Several reports have shown that the massive mortality of tilapia in Thailand is highly associated with sudden temperature changes during the day [16,17]. However, it was difficult to assess the impact of temperature on animal health and welfare in this study, as we only measured temperature once. We therefore suggest that the average all-day temperature of water should be reported, as previously described. Moreover, the addition of shaded areas may help reduce fluctuations in water temperature during the day [17].

Stocking densities are a major concern in semi-intensive and intensive tilapia farming practices. Overcrowded conditions may alter growth, reproduction, water quality, and fish behaviours [12,32,33]. In this study, we found negative correlations between fish density and water transparency, the presence of inhabitants, and fish behaviours, which indicates that tilapia raised in cage cultures may experience overcrowding conditions and exhibit inappropriate behaviours. Interestingly, we observed the co-cultivation of tilapia with white shrimps at farms A and E. The polyculture of tilapia and shrimp is common in many Southeast Asian countries, including Vietnam, Malaysia, the Philippines, and Thailand [34], and could potentially increase the biomass, growth, and survival rates of both species [35,36,37]. Similarly, the polyculture of tilapia and catfish at farm E was found to improve the profitability for the farmers, as reported in a previous study [38]. A further benefit of raising these two species together is the use of catfish to control the population of Nile tilapia in ponds [39]. We also observed that farms A and B regularly measured the water parameters using test kits and kept records, while the other farms only measured a few parameters, such as fish stocking or mortality, which were insufficient for evaluating welfare. It is challenging for Thai farmers to improve their operations by conducting systematic and regular water quality checks and maintaining farm management records.

### 4.3. Behavioural Indicators

Feeding and swimming behaviours are important indicators for assessing fish welfare during handling and captivity. When fish experience stress or discomfort, they may exhibit abnormal swimming behaviours or a reluctance to eat, which can negatively impact their growth performance, survival, and overall health and welfare over the long term. Our study revealed poor feeding behaviour scores for the tilapia at farms A and E, which could be attributed to factors such as the large pond sizes, overcrowding, and inadequate feeding methods. To overcome these challenges, identifying suitable and adequate feeding points based on pond sizes can ensure that fish have access to food throughout the entire culture area.

The harvesting process of tilapia in Thailand can have negative impacts on fish behaviours and health, as it often results in high stocking densities and poor welfare scores due to the fish being pushed into limited areas. The total capture of tilapia can also lead to overcrowding, which results in abnormal swimming behaviours and external lesions [12]. In the present study, fish handling during massive capture frequently involved prolonged periods of high stocking densities, which is considered unacceptable [9]. To mitigate these negative impacts, it may be necessary to decrease fish densities, use fish-sorting machines, or increase the number of workers to facilitate faster and more efficient fish handling and sorting during the capture process [12]. Additionally, knotless nets can be used as an alternative tool to reduce fish stress and injury during handling [40].

### 4.4. Nutrition Criteria

The evaluation of tilapia welfare relies heavily on nutritional criteria, which serve as indicators of farm efficiency and productivity. Our findings indicated that the majority of the farms included in the analysis provided fish feed with appropriate crude protein levels. Despite being raised for the same duration of 75 days, the difference in initial weights between farms F and G likely influenced the weight of the fish, even though the fish from F faced disease-related challenges. This difference in weights can be attributed to the varying initial weight of the fish, with an average ranging from 25 to 30 g in farm F, whereas farm G started with an average initial fish weight of 2 to 4 g. A well-balanced diet with sufficient protein is crucial for achieving optimal tilapia growth and survival, and a low FCR is indicative of efficient feed conversion. However, in this study, we observed that the fish raised in intensive farming practices, despite having suitable crude protein values, exhibited higher FCRs. This suggests that the stocking densities may have negatively affected the nutritional status of the tilapia, as previously reported [41,42]. An intriguing finding from our study was the co-cultivation of tilapia with white shrimp, as demonstrated at farms A and E. This co-cultivation approach appeared to promote a lower FCR and an increase in the growth rate of both species, as shown in earlier studies [35,36]. In addition, it is worth noting that all the farms in our study exhibited tilapia with higher K factor values than the standard values. This discrepancy may have arisen because the standard K factor values, which we derived from those for the supreme-strain tilapia in Brazil, may not be applicable to the Chitralada strain commonly cultured in Thailand [43,44]. Further standardisation of tilapia K factors should thus be addressed to improve the welfare assessments of tilapia in Thailand. 

### 4.5. Welfare Assessments in Thailand: Past, Present, and Future

Over the last decade, welfare assessments have become a significant issue in the aquaculture industry, and recognition of their importance among aquaculture stakeholders in Thailand is growing. As a result, there has been increasing emphasis on the development and implementation of effective welfare assessment and management practices. Many farms have begun to incorporate welfare assessments into their production systems with the aim of improving fish welfare and enhancing the sustainability of their operations. However, challenges still exist in the implementation of such practices. These challenges include a lack of standardised protocols, limited awareness and understanding of welfare issues among farmers and across the industry, and the need for further research and education in this area. In response, the Department of Fisheries has established the GAP, which focuses on several factors to ensure appropriate tilapia farm management [4]. Nevertheless, the welfare scoring system has not been incorporated into the standard to a satisfactory level, as reported in the case of other certifications [5,6,45]. To date, around 3000 of the 300,000 tilapia and red tilapia farms throughout Thailand have been certified against GAP standards [2]. In our study, only farms A and B had achieved GAP certification. It is thus essential to continue to develop and refine welfare assessment and management practices to ensure the long-term sustainability and success of the Thai aquaculture industry. 

It is important to underscore that the practical application of this initial protocol would enable Thai producers to be more attentive to fish health and farm management. However, several welfare indicators in this protocol, such as DO, transparency, alkalinity, and the TAN, differed from those of the Thai standard [4]. Moreover, some water quality parameters are evaluated in the Thai GAP standard but were not in our system. These include salinity, water hardness, and biological oxygen demand. It is noteworthy that welfare assessments for other fish species, such as salmon, rainbow trout, and pikeperch, include the weight score on each parameter, which enhances the interpretation of fish welfare at farms [8,9,10]. Additionally, Browning recently suggested welfare assessment improvements by increasing fish behavioural assessments as an outcome measurement of their protocol to better indicate the welfare status of fish [45]. Further improvements in welfare assessments in Thai tilapia farming should therefore be undertaken. Despite these challenges, our study could serve as the first step in a tilapia welfare assessment strategy in Thailand, as it prioritises critical points and leads to the implementation of corrective actions and the monitoring of the results as part of an ongoing welfare management programme. Another notable point is the potential of our protocol for adaptation into a mobile application, which may further encourage its on-farm adoption and use.

## 5. Conclusions

Based on our findings, the tilapia welfare assessment protocol adapted from the previous study [12] could be applied effectively across different farming practices, and its use by farm employees is feasible. Additionally, the developed protocol demonstrated a relative discriminative power, high on-field applicability, and a clear role in identifying critical tilapia welfare points, which may inform management decisions. Given the challenges associated with further protocol improvements, we believe that the presented format, which aligns with and resembles the welfare assessment protocols that have longer histories of usage and refinement for other species, may aid in determining the best future approaches. Lastly, protocol refinements are encouraged, for example, the integration of indicators into a single final score for each farm, the continuous refinement of the existing indicators, and the inclusion of new tilapia welfare indicators as they become available.

## Figures and Tables

**Figure 1 animals-13-02498-f001:**
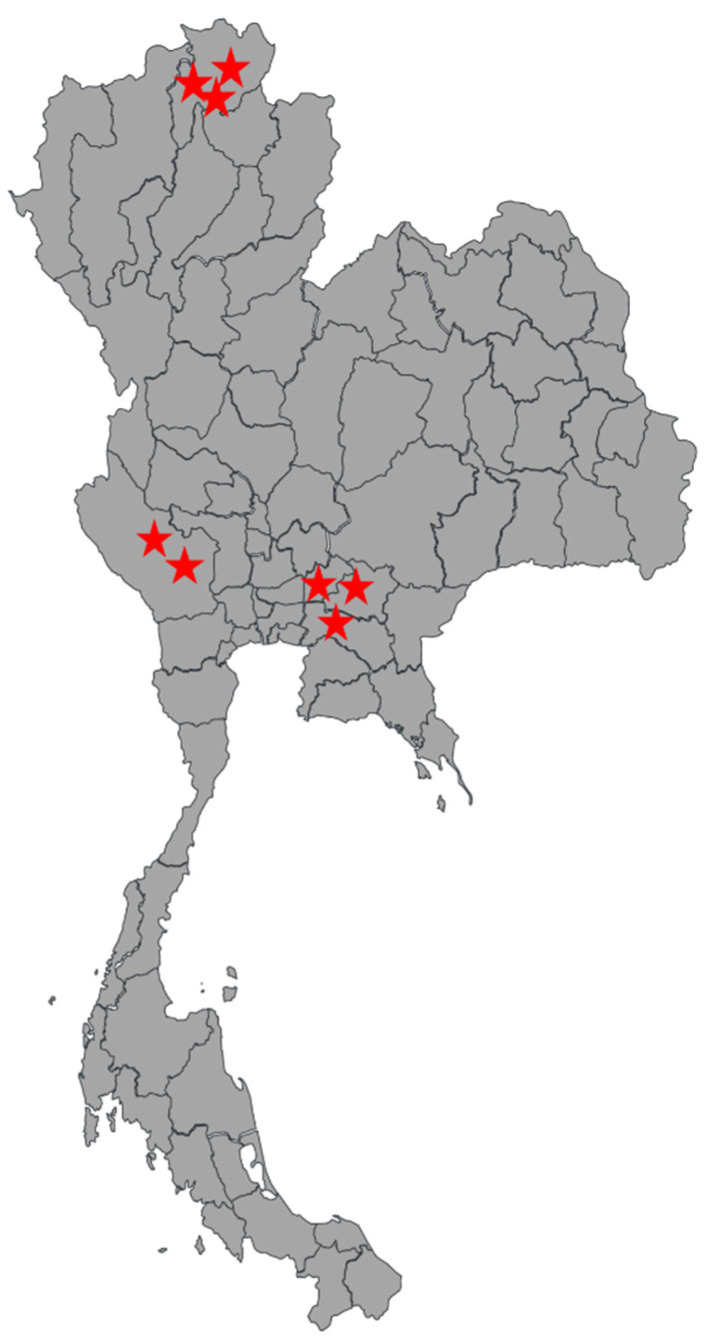
Location of the eight tilapia farms (indicated by stars) that were selected for the tilapia welfare assessments.

**Figure 2 animals-13-02498-f002:**
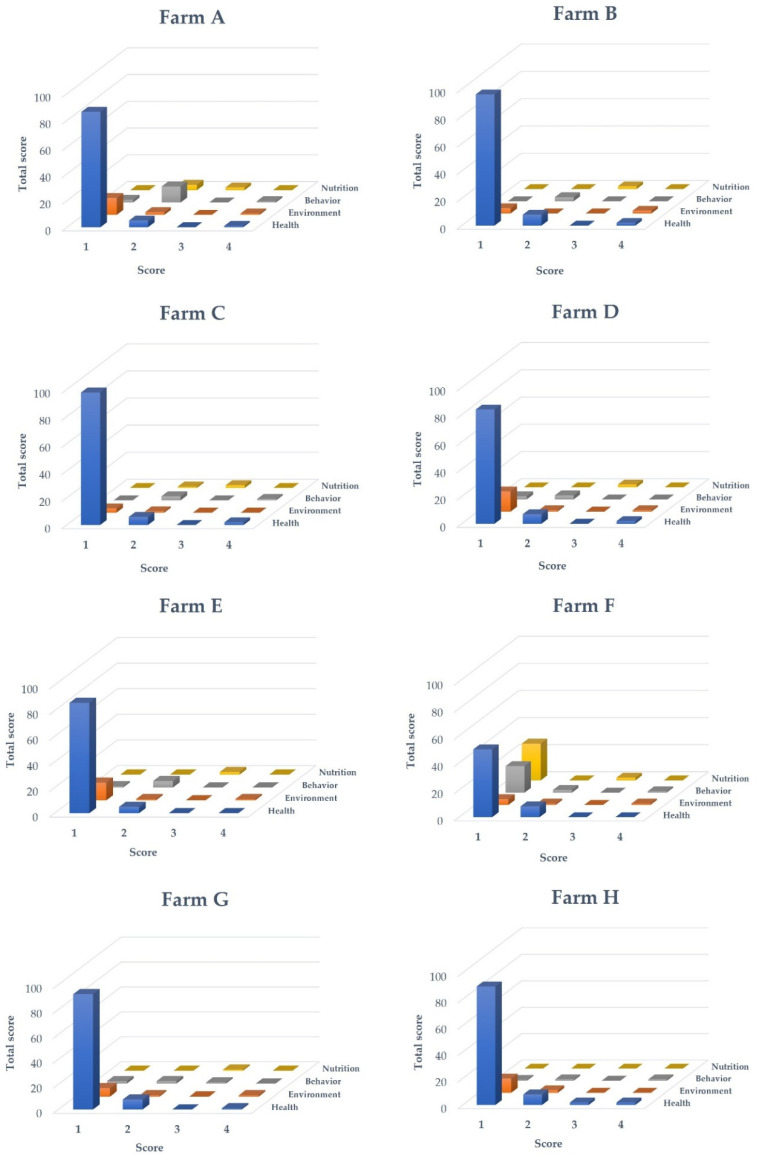
Overall scores for tilapia welfare at the eight farms measured in terms of the health (blue bar), environmental (orange bar), behavioural (grey bar), and nutritional (yellow bar) statuses of the fish.

**Figure 3 animals-13-02498-f003:**
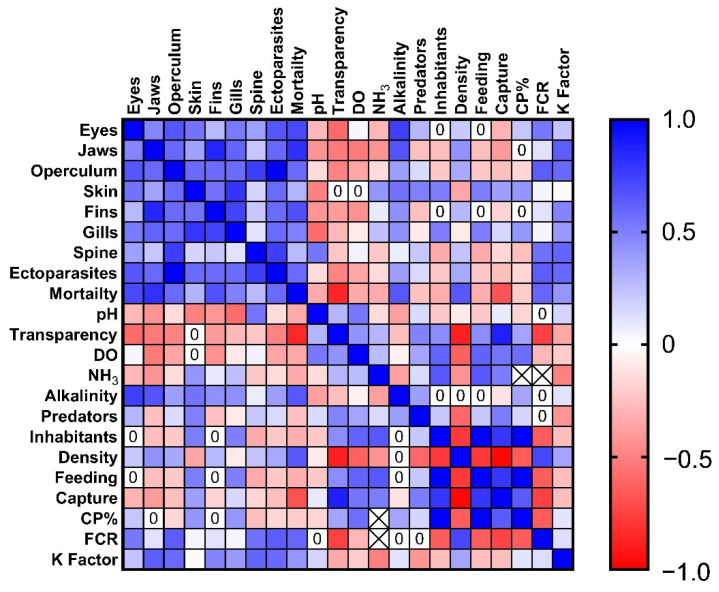
Heatmap showing the correlations between the welfare indicators. Spearman’s r value is illustrated as ranging from −1 to 1.

**Table 1 animals-13-02498-t001:** Demographic data of the eight farms.

Farm	Region	Culturing System	Species	Pond Size	Fish Weight (g) (Mean ± SD)	Fish Age (Days)
A	Eastern	Earthen pond	Nile tilapia, white shrimp	28,800 m^2^	710.6 ± 167.2	213
B	Northern	Earthen pond	Nile tilapia	3200 m^2^	306.2 ± 39.0	115
C	Northern	Earthen pond	Nile tilapia	4800 m^2^	151.5 ± 28.5	90
D	Northern	Earthen pond	Nile tilapia	4800 m^2^	508.6 ± 116.2	180
E	Central	Earthen pond	Nile tilapia, white shrimp, catfish	28,800 m^2^	246.9 ± 27.9	186
F	Western	Cage culture	Red tilapia	25 m^2^	292.0 ± 72.6	75
G	Western	Cage culture	Red tilapia	25 m^2^	106.6 ± 29.6	75
H	Eastern	Cage culture	Red tilapia	20 m^2^	542.8 ± 142.3	243

**Table 2 animals-13-02498-t002:** Score distribution (%) of the health indicators.

Raising System	Earthen Ponds																				Cages											
Health Indicators	Farm A (n = 43)				Farm B (n = 26)				Farm C (n = 45)				Farm D (n = 31)				Farm E (n = 30)				Farm F (n = 30)				Farm G (n = 30)				Farm H (n = 25)			
	1	2	3	4	1	2	3	4	1	2	3	4	1	2	3	4	1	2	3	4	1	2	3	4	1	2	3	4	1	2	3	4
Eyes	97.7	2.3	0	0	100	0	0	0	100	0	0	0	100	0	0	0	100	0	0	0	50	10	0	40	93.3	6.7	0	0	100	0	0	0
Jaws	95.3	4.7	0	–	100	0	0	–	100	0	0	–	90.3	9.7	0	–	100	0	0	–	50	0	50	–	96.7	3.3	0	–	92	4	0	–
Opercula	100	0	0	0	100	0	0	0	100	0	0	0	100	0	0	0	100	0	0	0	50	10	0	40	100	0	0	0	100	0	0	0
Skin	62.8	27.9	9.3	0	84.6	15.4	0	0	95.6	4.4	0	0	61.3	32.3	6.5	0	53.3	40	6.7	0	50	0	0	50	83.3	10	6.7	0	96	4	0	0
Fins	79.1	20.9	0	0	92.3	7.7	0	0	86.7	13.3	0	0	61.3	29	9.7	0	73.3	26.7	0	0	50	10	0	40	80	20	0	0	64	32	4	0
Gills	69.8	25.6	4.7	0	96.2	3.8	0	0	97.8	2.2	0	0	74.2	25.8	0	0	70	30	0	0	50	0	30	20	93.3	6.7	0	0	72	28	0	0
Spine	100	0	0	–	100	0	0	–	97.8	2.2	0	–	100	0	0	–	100	0	0	–	50	0	50	–	100	0	0	–	100	0	0	–
Ectoparasites	100	0	0	–	100	0	0	–	100	0	0	–	100	0	0	–	100	0	0	–	40	60	0	–	100	0	0	–	100	0	0	–
Mortality (%)	15				5				10				25				25				50				40				30			

**Table 3 animals-13-02498-t003:** The relative scores of the health indicators.

Raising System	Relative Score							
Health Indicators	Earthen Ponds					Cages		
	Farm A (n = 43)	Farm B (n = 26)	Farm C (n = 45)	Farm D (n = 31)	Farm E (n = 30)	Farm F * (n = 30)	Farm G (n = 30)	Farm H (n = 25)
Eyes ^a^	1.02	1.00	1.00	1.00	1.00	2.30	1.07	1.00
Jaws ^a^	1.05	1.00	1.00	1.10	1.00	2.00	1.03	1.08
Opercula ^a^	1.00	1.00	1.00	1.00	1.00	2.40	1.00	1.00
Skin ^b^	1.47	1.15	1.04	1.45	1.53	2.50	1.23	1.04
Fins ^b^	1.21	1.08	1.13	1.48	1.27	2.30	1.20	1.40
Gills ^a^	1.35	1.04	1.02	1.26	1.30	2.20	1.07	1.28
Spine ^a^	1.00	1.00	1.02	1.00	1.00	2.00	1.00	1.00
Ectoparasites ^a^	1.00	1.00	1.00	1.00	1.00	1.60	1.00	1.00
Mortality ^c^	2	1	1	2	2	3	3	2

* Indicates *p*-value < 0.05 compared with other farms (using Dunnett’s); ^a, b, c^ indicate statistically significant differences between health indicators.

**Table 4 animals-13-02498-t004:** Environmental indicators and scores.

Raising System	Earthen Ponds										Cages					
Environmental Indicator	Farm A		Farm B		Farm C		Farm D		Farm E		Farm F		Farm G		Farm H	
	Value	Score	Value	Score	Value	Score	Value	Score	Value	Score	Value	Score	Value	Score	Value	Score
Temperature (°C)	28.4	1	28.4	1	30.0	1	30.4	1	30.0	1	27.0	1	27.8	1	28.2	1
pH	8.5	1	8.0	1	8.7	2	7.7	1	6.8	1	7.5	1	7.5	1	7.2	1
Transparency (cm)	20	3	21	3	18	3	24	3	15	3	35	1	30	1	25	1
Dissolved oxygen (mg/L)	8.25	4	5.47	1	8.9	4	6.39	1	3.4	3	6.16	1	5.2	2	5.38	1
NH_3_ (mg/L)	0.00	1	0.00	1	0.00	1	0.01	1	0.08	2	0.00	1	0.00	1	0.00	1
NO_2_^−^ (mg/L)	0.00	1	0.00	1	0.00	1	0.10	1	0.00	1	0.00	1	0.00	1	0.00	1
Alkalinity (mg/L of CaCO_3_)	102	2	85	1	34	1	102	2	85	1	119	2	119	2	51	1
Shading (%)	0	3	0	3	0	3	0	3	0	3	0	3	0	3	60	3
Predators	UP	3	UP	3	UP	3	UP	3	UP	3	UP	3	UP	3	CP	2
Inhabitants	UP	3	A	1	A	1	A	1	UP	3	A	1	A	1	A	1
Density (fish/m^2^)	1.25	1	3.13	1	3.13	1	3.13	1	2.08	1	60.0	1	60.0	1	90.0	1

A, absence; CP, controlled presence; UP, uncontrolled presence.

**Table 5 animals-13-02498-t005:** Behavioural assessment and scores.

Raising System	Earthen Ponds					Cage		
Behavioural Indicators	Farm A	Farm B	Farm C	Farm D	Farm E	Farm F	Farm G	Farm H
	Score	Score	Score	Score	Score	Score	Score	Score
Feeding	4	1	1	1	4	1	1	1
Capture	4	4	4	4	4	4	4	1

**Table 6 animals-13-02498-t006:** Nutrition indicators and scores.

Raising System	Earthen Ponds										Cages					
Nutritional Indicators	Farm A		Farm B		Farm C		Farm D		Farm E		Farm F		Farm G		Farm H	
	Value	Score	Value	Score	Value	Score	Value	Score	Value	Score	Value	Score	Value	Score	Value	Score
Fish weight (g) (mean ± SD)	710.6 ± 167.2	–	306.2 ± 39.0	–	151.5 ± 28.5	–	508.6 ± 116.2	–	246.9 ± 27.9	–	292.0 ± 72.6	–	106.6 ± 29.6	–	542.8 ± 142.3	–
Fish age (days)	213	–	115	–	90	–	180	–	186	–	75	–	75	–	243	–
Use commercial feed	Yes	–	Yes	–	Yes	–	Yes	–	No	–	Yes	–	Yes	–	Yes	–
Crude protein ratio (CP%)	25	2	30	1	30	1	30	1	ND	–	30	1	32	1	30	1
Feed conversion ratio (FCR)	1.1	1	1.3	1	1.3	1	1.2	1	ND	–	1.55	2	1.41	2	1.3	1
K factor (mean ± SD)	2.53 ± 0.27	3	2.22 ± 0.20	2	2.53 ± 0.21	3	2.26 ± 0.21	2	2.20 ± 0.17	2	3.7 ± 1.44	3	2.2 ± 0.34	2	2.75 ± 0.62	3

## Data Availability

The data presented in this study are available on request from the corresponding author. The data are not publicly available due to the anonymity granted to all participating parties.

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
