# Peer review of "Assessment of Tilapia (Oreochromis spp.) Welfare in the Semi-Intensive and Intensive Culture Systems in Thailand"

_animals, 2023, doi:10.3390/ani13152498_

Round 1

Reviewer 1 Report

I would like to congratulate the authors on the research done and the manuscript prepared. The article is interesting and well-written. I have added my comments in the attached file. Additionally, in Table S2 of the supplementary material, it seems to me that the middle range of values for transparency should be up to 65 (it is 69). I also have a question about the sample size form a given fish farm - how was it determined?

Author Response

Dear Reviewer 1,

We are appreciated for your valuable comments on our manuscript entitled “Assessment of tilapia (Oreochromis spp.) welfare in the semi-intensive and intensive culture systems in Thailand". All suggestions have been amended and proofread. We hope our latest manuscript is completed and ready for the publications.

Yours sincerely,

Tuchakorn Lertwanakarn,

(On behalf of all authors)

Reviewer 2 Report

Overall recommendations/comments:

This is an interesting paper and gives information about welfare indicators in tilapia. Authors propose an interesting welfare assessment protocol based on health, environmental, behavioral and nutrition indicators, to use in tilapia production systems. This study shows promising results and can be applicated to the aquaculture industry. The work supplements several other investigations covering the same topic and many of these papers appear in the references. On the whole, the paper is well-written and considerable thought has been put into interpreting the reported findings. The structure and format are good.

Please define the name of each fish species the first time that each one appears on the manuscript.

I think the work covered in the manuscript is appropriate for publication in Animals Journal after some major changes.

Abstract:

In the abstract, there is missing information that is necessary to understand the manuscript. The authors mention that they have observed differences in various parameters, but they do not precisely define which groups were different from each other. I understand that the word limit may have influenced this, but it is necessary to include, at the very least, the most notable results of the study. Furthermore, the conclusion is very broad, so I recommend that the authors define the main conclusion of their study in a more specific manner.

Introduction:

The introduction is good, emphasizing the principal subjects of the paper to present the hypothesis.

Material and methods:

The "Material and Methods" section is good and allows other authors to replicate the experiment.

The experimental design includes comparisons between farms with earthen ponds and farms with cages, but in the former, the cultivated species were Nile tilapia, while in the latter, they were red tilapia. This could be a problem when comparing the data from the two systems. Do the authors believe that having different species could affect the results/conclusions obtained in these comparisons?

Results:

The main problem with the manuscript being accepted is in the results section. Some comments:

It would be necessary to include the meaning of the scores (1, 2, 3, 4) for each parameter studied in the tables as table footnotes. The authors cite the manuscript on which they have based the development of this welfare protocol and include it as supplementary material, but the tables presented in the text should be able to be understood on their own.

Table 3: Please include the meaning of letters on health indicators as table footnotes.

In section 3.2.2., there is no table referenced for environmental indicators data.

In section 3.2.3., the authors refer to Table 4 for behavioral indicators data, but Table 4 is about environmental indicators. I think that the authors forgot to include the table about behavioral indicators, so it is impossible to review that part of the results.

I think that the final paragraph of section 3.2.4 (Nutritional indicators) could be placed at the beginning of the results section because this paragraph speaks about statistical comparisons between farms and between production systems (earthen ponds and cages). The authors also reported no significant differences in the overall welfare scores across all the farms but with a p = 0.01. In the statistical analysis section in Material and Methods, they defined p-values < 0.05 as significant, so a p = 0.01 would be considered significant.

Discussion and conclusion:

The discussion is good, and the authors show that they know previous research about the subject they are studying.

Author Response

Dear Reviewer,

We are appreciated on your valuable comments on our manuscript entitled “Assessment of tilapia (Oreochromis spp.) welfare in the semi-intensive and intensive culture systems in Thailand”. All suggestions have been amended and proofread. All suggestions have been amended and proofread. We hope our latest manuscript is completed and ready for the publications.

Yours sincerely,

Tuchakorn Lertwanakarn

(On behalf of all authors)

Round 2

Reviewer 1 Report

The authors addressed all suggestions and revised the manuscript accordingly

Reviewer 2 Report

All suggestions have been amended and now I consider the manuscript is ready to publish.